# Material-agnostic machine learning approach enables high relative density in powder bed fusion products

Jaemin Wang ®[1], Sang Guk Jeong ®[1], Eun Seong Kim[1], Hyoung Seop Kim ®[2,3,4,5] & Byeong-Joo Lee ®[1] ✉

This study introduces a method that is applicable across various powder materials to predict process conditions that yield a product with a relative density greater than 98% by laser powder bed fusion. We develop an XGBoost model using a dataset comprising material properties of powder and process conditions, and its output, relative density, undergoes a transformation using a sigmoid function to increase accuracy. We deeply examine the relationships between input features and the target value using Shapley additive explanations. Experimental validation with stainless steel 316 L, AlSi10Mg, and Fe60Co15Ni15Cr10 medium entropy alloy powders verifies the method's reproducibility and transferability. This research contributes to laser powder bed fusion additive manufacturing by offering a universally applicable strategy to optimize process conditions.

Laser powder bed fusion (L-PBF), also known as selective laser melting, is a high-precision additive manufacturing (AM) method to produce metal components that have superior mechanical properties, and that can use a wider range of metal feedstock materials than other AM methods[1]. During L-PBF, a layer of powdered metal is spread on a bed, then a laser is used to fuse the powder layer. The molten layer is allowed to cool rapidly, then another thin layer of powder is spread over it. This process is repeated until the desired part is fully constructed layer-by-layer. This process permits creation of intricate structures with high-quality material properties.

The material properties of parts created using L-PBF are heavily influenced by process parameters, such as laser power $P$ [W], scan speed $v$ [mm/s], hatch distance $h$ [mm], and layer thickness $t$ [mm]. Inappropriate combinations of these parameters can lead to keyhole formation or lack of fusion, both of which result in parts that have undesirable porosity and microstructure[2]. Pores can provide initiation sites for cracks within the structure of AM-produced parts[3], so high porosity within the material can significantly compromise its

mechanical properties. Consequently, effective L-PBF requires use of process parameters that yield parts with low porosity, i.e., high relative density $\tilde{\rho}$. $\tilde{\rho}$ depends on laser energy density $E_d$ [J/mm³], calculated as[4]

$$E_d = \frac{P}{v \cdot h \cdot t} \tag{1}$$

Pores can occur due to lack of fusion occur when $E_d$ is insufficient, or due to vaporization when $E_d$ is excessive. However, the optimal $E_d$ to achieve high $\tilde{\rho}$ varies among materials, and the $\tilde{\rho}$ of parts fabricated using the same material and $E_d$ can differ depending on the process parameters used[5] (Fig. 1). Because of this variability, the parameters must be tailored to the material to ensure the best possible results.

Numerous prior studies[6–14] have used ML as a tool to predict and optimize part quality, such as $\tilde{\rho}$ by using process conditions to predict various properties of products. However, these studies have considered only one powder material for training, prediction, and model validation.

[1]Department of Materials Science and Engineering, Pohang University of Science and Technology (POSTECH), Pohang 37673, Republic of Korea. [2]Graduate Institute of Ferrous and Eco Materials Technology (GIFT), Pohang University of Science and Technology (POSTECH), Pohang 37673, Republic of Korea. [3]Center for Heterogenic Metal Additive Manufacturing, Pohang University of Science and Technology (POSTECH), Pohang 37673, Republic of Korea. [4]Institute for Convergence Research and Education in Advanced Technology, Yonsei University, Seoul 03722, Republic of Korea. [5]Advanced Institute for Materials Research (WPI-AIMR), Tohoku University, Sendai 980-8577, Japan. ✉e-mail: calphad@postech.ac.kr

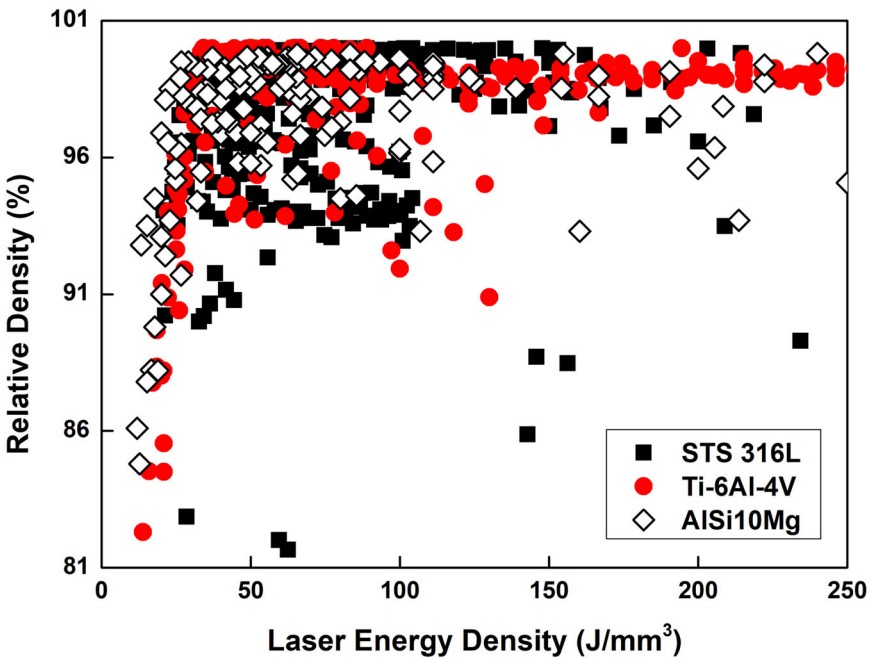

**Fig. 1 | Scatter plot of relative density [%] vs laser energy density [J/mm³] for STS 316 L, Ti-6Al-4V, and AlSi₁₀Mg powders in L-PBF.** Description: The scatter plot highlights the variation in optimal energy density required to achieve high relative density for each material, as well as the differing relative densities achieved with the same energy density when different process parameters are used. The references of the data are given in the supplementary materials. Source data are provided as a Source Data file.

On the other hand, Cacace & Semeraro[15] presented an intriguing approach to calculate the lack of fusion probability by integrating semi-analytical thermal model results with a geometric-based defect model. Their research offered an innovative method of establishing PBF process conditions for a variety of materials, with a focus on balancing porosity and productivity.

Despite the strengths of their approach, there are limitations in terms of time-efficiency and adaptability. Obtaining process conditions via their method can be time-consuming, and new regression models for melt pool depth and width are required for each material. Moreover, although $E_d$ has been suggested as a simple criterion for calculating optimal process parameters across different materials, it has proven insufficiently reliable for this purpose[5].

Our research aims to overcome these challenges by developing an ML model that optimizes $\tilde{\rho}$ simply, time-efficiently, and regardless of the material used. Our goal is to create an approach that enhances both accuracy and applicability across a wide array of materials.

In the present study, we seek to establish an ML model as a standard, aiming to replace $E_d$ and the work of Cacace & Semeraro. To accomplish this goal, we trained a model that can determine whether the $\tilde{\rho}$ of a product is sufficiently high, when manufactured under specific process conditions using a particular powder. The model is trained using the material properties of the powder and the process parameters. We also developed a method that uses the model to suggest process parameters to achieve high $\tilde{\rho}$ in a product. To prove the reliability of the model and method, we predicted the optimal process conditions for STS 316 L, AlSi10Mg, and Fe60Co15Ni15Cr10 MEA, and experimentally verified the predicted process conditions.

This study presents three contributions. First, our method can propose optimized process parameters for any material, provided that its physical properties are known. This material-agnostic approach increases its applicability. Second, we modified the target values to facilitate the training of the ML model, which in turn improved its accuracy. Lastly, by using explainable AI (XAI) on the trained model, we analyzed the correlations among material properties, process parameters, and $\tilde{\rho}$, and obtained valuable insights into the relationships that affect these factors.

In this paper, we calculate $\tilde{\rho}$ as 100% minus area fraction of porosities, as measured using image analysis, aligning with the approach adopted in previous studies[16–18]. The decision to use image analysis over the Archimedes' method was made primarily due to the potential inaccuracies associated with the latter. The Archimedes' method calculates $\tilde{\rho}$ by dividing the product's density, as measured by this method, by the ideal density inferred from the powder's composition. However, this approach can lead to incorrect measurements of $\tilde{\rho}$ due to the evaporation of low-melting-point elements during product manufacturing, which can subsequently alter the composition. Given our focus on the porosity itself, we found image analysis, which directly relates to porosity, to be the more appropriate method for measuring $\tilde{\rho}$.

## Results and discussion
### Performance assessment of the developed ML model
To demonstrate that the present model is more accurate than both the simple regression model and the simple classification model, we compared their performance parameters, i.e., recall, precision, and accuracy (Table 1). Recall represents the proportion of positive instances that the model identifies out of all actual positive instances. Precision quantifies the proportion of true positive instances among all instances predicted as positive by the model. Accuracy represents the proportion of correct predictions made by the model, considering

**Table 1 | Comparison of recall, precision, and accuracy [all %] for the simple regression model, the simple classification model, and the present model**

| Criterion | Model | | |
|---|---|---|---|
| | Simple regression | Simple classification | Present |
| Recall | 81.8 | 89.3 | 88.2 |
| Precision | 91.0 | 85.2 | 89.3 |
| Accuracy | 84.7 | 84.7 | 87.0 |

both true positive and true negative instances in relation to the total number of instances. The models were implemented using XGBoost[19].

The present model is a regression model that uses a sigmoid function to transform the target value, then predicts the transformed value. In contrast, the simple regression model predicts the target value directly, and the simple classification model predicts the target value that is transformed by binary encoding.

The formula for transforming the target value in the present model is

$$y_{present} = \frac{1}{1 + e^{(y_{original} - 98)}}, \tag{2}$$

where $y_{present}$ represents the target value, and $y_{original}$ denotes the original target value. A $\tilde{\rho}$ of 98% is high enough to ensure that product material properties are not severely compromised[20–22], so we established 98% as our criterion for high $\tilde{\rho}$. When the $\tilde{\rho}$, which serves as the target value, is transformed using Eq. 2, $\tilde{\rho} \geq 98\%$ yield values between 0.5 and 1, and $\tilde{\rho} < 98\%$ result in values between 0 and 0.5. In the present model, the predicted value is converted to 1 if it exceeds 0.5 and to 0 if it does not when evaluating recall, precision, and accuracy. For the regression model, the predicted value is transformed to 1 when it surpasses 98% and to 0 when it falls below 98% for the purpose of evaluating recall, precision, and accuracy.

The three models had differing strengths (Table 1). The simple regression model had the highest precision, and the simple classification model had the highest recall. The present model had higher accuracy than the other two models, but only moderately high precision and recall.

The simple regression model's strength in precision stems from its tendency to underestimate the predicted value when dealing with data that have $\tilde{\rho} \geq 98\%$. To substantiate this observation, we compared the averaged raw value and averaged prediction of two data sets: one with a $\tilde{\rho}$ below the average and another with a $\tilde{\rho}$ above the average. The average $\tilde{\rho}$ of all data sets used was 96.17%. The below-average data set had a mean $\tilde{\rho} = 88.4\%$, and a predicted mean $\tilde{\rho} = 88.9\%$. In contrast, the above-average dataset had a mean $\tilde{\rho} = 98.8\%$, and a predicted mean $\tilde{\rho} = 98.6\%$.

The simple regression model overestimated data that had $\tilde{\rho}$ below average, but underestimated data that had $\tilde{\rho}$ above average. This inclination to underestimate above-average $\tilde{\rho}$ increases the likelihood that the model will judge data with actual $\tilde{\rho} \geq 98\%$ as being <98%. Consequently, this tendency results in a decrease in recall and an increase in precision.

The simple classification model obtained results opposite to those of the simple regression model due to the classification model's method of converting the target value to 1 when $\tilde{\rho} \geq 98\%$ and 0 otherwise. With 57.78% of the data having $\tilde{\rho} \geq 98\%$, so more target values were 1 than 0. This difference in frequency causes the model to favor a judgment of 1, because it is more likely to be correct than a judgement of 0. In fact, the proportion of data predicted with $\tilde{\rho} \geq 98\%$ in the simple classification model was 58.7%, confirming that the predictions are biased, leading to lower precision and higher recall.

However, the present model mitigates these flaws by using a sigmoid function to normalize the target. The sigmoid-converted had an average $\tilde{\rho} = 0.5019$. Because the model is technically a regression model, the problem of underestimating values when they are $\geq 0.5019$ and overestimating them when they are <0.5019 persists. The dataset with above-average $\tilde{\rho}$ had average $\tilde{\rho} = 0.7654$, and a predicted average $\tilde{\rho} = 0.7369$. The dataset with below-average $\tilde{\rho}$ had average $\tilde{\rho} = 0.1445$, and a predicted average $\tilde{\rho} = 0.1864$.

However, the average $\tilde{\rho}$ of the entire dataset is close to the classification point of 0.5, so slight underestimations or overestimations do not significantly affect the classification result. This result occurs because not all data with $\tilde{\rho} \geq 98\%$ are underestimated, unlike the

results of the regression model. In addition, the present model is not strictly a classification model, so it does not suffer from bias due to data frequency. Consequently, the present model achieves higher accuracy than the other two models, despite only moderately high recall and precision.

Furthermore, the reason that the present model is more accurate than the simple regression model and the simple classification model can be explained as follows. The simple regression model is somewhat inefficient to minimize the prediction error per training iteration to precisely match the process conditions that have $\tilde{\rho} \geq 99.5\%$ or $\leq 96\%$, when accurate matching of values is not necessary. The simple classification model has a different problem: for example, $\tilde{\rho} = 97.9999$ is classified as 0, but $\tilde{\rho} = 98$ is classified as 1, despite the negligible difference of 0.0001. In addition, both 100 and 98 are classified as 1, whereas both 97 and 80 are classified as 0. This binning into 1 or 0 loses information or trends that could be gleaned from the true values of $\tilde{\rho}$. This loss reduces the model's accuracy and diminishes the accuracy when analyzing the model. Hence, the present model effectively overcomes these limitations of the other models and is therefore more reliable than they are for predicting $\tilde{\rho}$.

## SHAP analysis of the input features

Shapley additive explanations (SHAP)[23,24] analysis was conducted to uncover the correlations among process parameters ($E_d$, $P$, $v$, $h$, $t$), material properties of powder (thermal conductivity $k$ [W/(m·K)], material density $\rho_m$ [g/cm³], melting point $T_M$ [°C], reflectivity $R$ [%], and specific heat capacity $C_p$ [J/(g·K)]), and $\tilde{\rho}$. The SHAP score quantifies the extent and direction of each feature's contribution to the model's prediction. The SHAP score enables detection of whether a particular feature increases or decreases the predicted value and the magnitude of its effect. The average of the absolute values of the SHAP score quantifies the effect of each feature on the model's prediction; i.e., the importance of a given feature within the model.

The input features in the present model were composed of process parameters and material parameters of the powder. SHAP analysis (Fig. 2a) ranked the importance of process parameters as $v > P > t > h$, and the importance of material properties as $k > \rho_m > T_M > R > C_p$. Although the degree of importance varied among the features, this analysis indicates that they all contribute to the prediction to some extent.

The SHAP score can be divided into a main effect and an interaction effect. A main effect represents the isolated contribution of a single feature to the prediction, and signifies the individual impact of that feature. An interaction effect accounts for the combined influence of two features on the prediction, highlighting the contribution resulting from their interaction.

The contribution of each feature's main effect on the model's prediction $\tilde{\rho}$ was visualized by calculating the SHAP main values (Fig. 2b–k). The overall trend of these SHAP main values is represented by lines calculated using locally weighted regression, to clarify the tendencies.

Each process parameter had a distinct effect on $\tilde{\rho}$. When $E_d$ was less than ~50 J/mm³, $\tilde{\rho}$ decreased significantly (Fig. 2b), possibly because energy delivered to the powder was insufficient to complete its fusion. When scan speed was excessively high or excessively low $\tilde{\rho}$ decreased (Fig. 2c), possibly because inappropriate scan speed leads to increased porosity caused by lack of fusion or vaporization. High $P$ increased $\tilde{\rho}$ (Fig. 2d); at very low $P$, $\tilde{\rho}$ dropped substantially due to lack of fusion. Increasing $t$ reduced predicted $\tilde{\rho}$ (Fig. 2i), because powders below the laser-irradiated area may not receive enough thermal energy melt. $h$ reduced $\tilde{\rho}$ if $h$ is too small or too large (Fig. 2j); we speculate that too-small $h$ causes heat concentration and keyhole formation, whereas too-large $h$ results in lack of fusion and increased porosity due to unmelted areas between scan tracks.

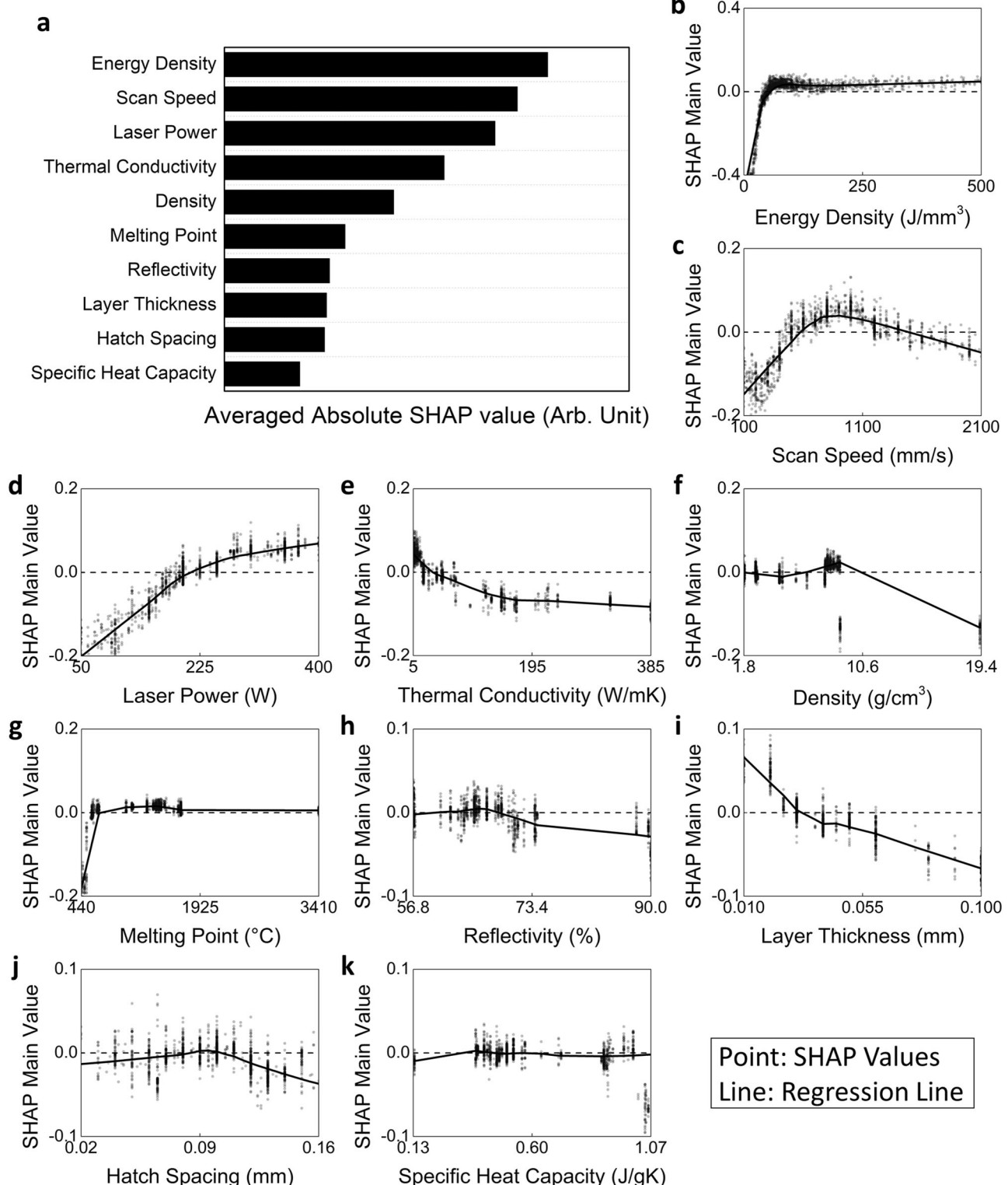

**Fig. 2 | Feature importance and SHAP analysis of input features without considering interactions.** Description: **a** Average absolute SHAP scores of input features, highlighting the relative importance of process parameters and material properties of powder. Visualization of the SHAP main scores for **b** energy density, **c** scan speed, **d** laser power, **e** thermal conductivity, **f** density, **g** melting point, **h** reflectivity, **i** layer thickness, **j** hatch spacing, and **k** specific heat capacity and their effects on the model's output, relative density. Source data are provided as a Source Data file.

Material parameters also had distinct effects. As $k$ decreased, $\widetilde{\rho}$ increased (Fig. 2e). Low $k$ means that thermal energy remains concentrated in the laser-irradiated area, so melting can be homogeneous. Increase in $k$ causes increase in the $E_d$ that is required to achieve homogeneous melting[25]. $\rho_m$ had a threshold effect (Fig. 2f): above a

certain level it reduced the predicted $\widetilde{\rho}$. This result implies that $\rho_m$ affects the model's prediction by interactions with other inputs; this possibility will be analyzed later. $T_M$ did not seem to have a significant effect on the predicted $\widetilde{\rho}$ (Fig. 2g), except that a very low $T_M$ did decrease the predicted $\widetilde{\rho}$. Materials that have low $T_M$ also have low boiling points, so they

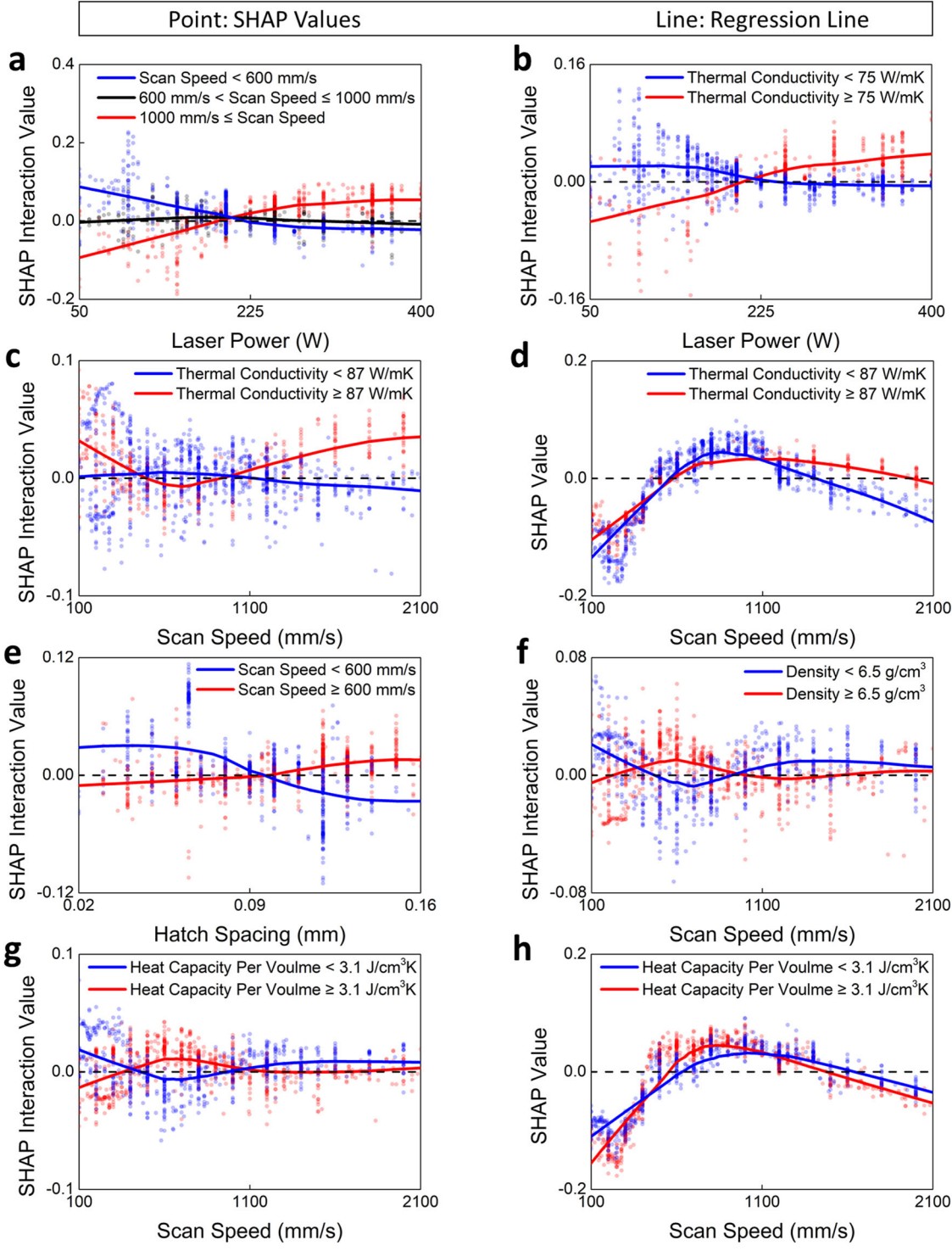

**Fig. 3 | SHAP analysis of the interactions between input features.** Description: Visualization of the SHAP interaction values for **a** interaction between laser power and scan speed, **b** interaction between laser power and thermal conductivity, **c** interaction between scan speed and thermal conductivity, **e** interaction between hatch spacing and scan speed, **f** interaction between density and scan speed, and **g** interaction between heat capacity per volume and scan speed. **d** Visualization of the combined SHAP scores, representing the sum of SHAP interaction scores for scan speed and thermal conductivity, and the SHAP main scores for scan speed. **h** Visualization of the combined SHAP scores, representing the sum of SHAP interaction scores for scan speed and heat capacity per volume, and the SHAP main scores for scan speed. Source data are provided as a Source Data file.

are likely to vaporize; this process causes keyhole formation and reduced $\tilde{\rho}$. Increase in $R$ decreased the predicted $\tilde{\rho}$ (Fig. 2h), because high $R$ reduces energy transfer from the laser to the powder, so homogeneous melting is difficult to achieve. $C_p$ also showed a threshold effect (Fig. 2k): above a certain level it reduced the predicted $\tilde{\rho}$. The effect of $C_p$ is influenced by interactions with other inputs, and will be analyzed later.

## SHAP analysis of interactions between input features
The contribution of input feature interactions to the model's prediction was visualized by calculating the SHAP interaction values (Fig. 3). All computed SHAP interaction values are grouped based on whether one of the input features used in each interaction calculation has a high or low value. To clearly identify the trends within each group, a line to

**Table 2 | Interactions between input features ranked by their averaged absolute SHAP values**

|     | Interaction between input features | Average absolute SHAP score |
| --- | --- | --- |
| #1 | Laser Power – Scan Speed | 26.82 |
| #2 | Laser Power – Thermal Conductivity | 16.48 |
| #3 | Scan Speed – Thermal Conductivity | 15.67 |
| #4 | Hatch Spacing – Scan Speed | 14.27 |
| #5 | Scan Speed – Density | 11.59 |
| #6 | Scan Speed – Reflectivity | 10.54 |
|     | ... |  |
| #36 | Melting Point – Specific Heat Capacity | 1.96 |

represent the tendency of the SHAP interaction scores was calculated using locally weighted regression.

The SHAP interaction values between $P$ and $v$ are illustrated in Fig. 3a. The SHAP interaction score was negative when low $P$ was combined with high $v$; i.e., this interaction decreases predicted $\tilde{\rho}$. A lack-of-fusion phenomenon, which leads to a drop in $\tilde{\rho}$, occurs under these conditions. However, the SHAP interaction score was also negative when high $P$ was combined with low $v$; i.e., this interaction also decreases predicted $\tilde{\rho}$. Under such conditions, keyhole formation, which reduces $\tilde{\rho}$, occurs. When $P$ and $v$ are appropriately balanced, their interaction increases $\tilde{\rho}$.

This analysis demonstrates that the trained ML model effectively recognizes the lack-of-fusion and keyhole-formation phenomena, which cause a drop in $\tilde{\rho}$. Consequently, the model's deep understanding of the processes occurring in L-PBF enables insights through this analysis of interactions between input features. However, not all SHAP interaction scores exhibit a clear correlation. Only relationships between two features can be analyzed, so changes in the SHAP interaction score due to the influence of other features cannot be fully explained.

The interactions between input features and their corresponding averaged absolute SHAP scores were sorted in descending order (Table 2). Interaction analyses were performed only for interactions that had average absolute SHAP interaction scores ranking in the top 5. Interactions that involved $E_d$, a dependent input feature determined by other input features, are not displayed in Table 2 and are not included in the analysis.

The SHAP interaction values between $k$ and $P$ are displayed in Fig. 3b. For materials with high $k$, increasing $P$ contributes to an increase in $\tilde{\rho}$, because a material with high $k$ demands a higher $P$ to achieve homogeneous melting, as heat dissipates more rapidly into the surroundings than with materials of low $k$ when subjected to the same $P$. In contrast, for materials with low $k$, use of a low $P$ increased the predicted $\tilde{\rho}$, possibly because adequate heat energy for melting is attained even at low $P$, because heat is retained near the irradiation site.

$k$ and $v$ showed a complex interaction (Fig. 3c), which is challenging to analyze using only the SHAP interaction scores. Consequently, an analysis was conducted using the sum of the SHAP interaction score for both features and the SHAP main score for $v$ ('SHAP Value', Fig. 3d). For materials with high $k$, the range of appropriate $v$ to increase $\tilde{\rho}$ is broader than the range for the materials with low $k$. In materials with low $k$, heat remains concentrated in the laser-irradiated area, and we suggest that lack of fusion may occur if $v$ is too fast, and vaporization may occur if $v$ is too slow. In contrast, materials that have high $k$ are less prone to these effects, and therefore less sensitive to variation in $v$, than materials that have low $k$.

The SHAP interaction values between $v$ and $h$ are presented in Fig. 3e. At high $v$, increasing the $h$ increased $\tilde{\rho}$, whereas at low $v$, reducing the $h$ increased $\tilde{\rho}$. This finding contradicts the $E_d$ equation

(Eq. 1). If a certain material allows for a range of $E_d$, then reducing $h$ to stay within that range would increase $\tilde{\rho}$ as $v$ rises. However, the SHAP interaction value analysis suggests that the $\tilde{\rho}$ would be lowered in this case.

The reason for this discrepancy is that the SHAP score represents the contribution of the corresponding input feature or interaction to each predicted value. If both $v$ and $h$ are excessively large or small, $E_d$ will be outside the appropriate range and will contribute to a significant decrease in $\tilde{\rho}$. For instance, when STS 316 L has process conditions with $h = 0.15$ mm and $v \geq 1200$ mm/s, the $E_d$ is outside the appropriate range, so the SHAP score of the $E_d$ parameter rapidly decreases (Supplementary Fig 1).

As long as $E_d$ remains within an appropriate range, increasing both $v$ and $h$ can increase $\tilde{\rho}$. In situations where $P$ must be increased, increasing only one of $v$ and $h$ to maintain an adequate $E_d$ may result in a slight decrease in $\tilde{\rho}$.

To better understand the interaction between $v$ and $\rho_m$, a new input feature was created as heat capacity per volume $C_p \cdot \rho_m = C_v$ [J/(K•cm$^3$)] and a new model was trained with $C_v$ and without $C_p$ or $\rho_m$. Use of $C_v$ and removal of $C_p$ and $\rho_m$ led to a slight loss of information represented by each input feature, so the model's accuracy decreased to 85.6%. Furthermore, the complex trend of the SHAP interaction between $v$ and $C_v$ impedes the identification of the interaction by using only the SHAP interaction score. Therefore, analysis was conducted using the SHAP value obtained by adding the SHAP interaction value between $v$ and $C_v$, and the SHAP main score of $v$.

The interactions between $v$ and $\rho_m$ (Fig. 3f) and between $v$ and $C_v$ (Fig. 3g) showed similar tendencies. This observation suggests that in the present model, $\rho_m$ rather than $C_p$ could be representing $C_v$. This effect can occur because $\rho_m$ has a larger normalized distribution width than $C_p$. Variation in $C_p$ across the materials in the dataset is relatively minor, so the suggestion that $C_v$ is predominantly determined by $\rho_m$ is plausible.

The SHAP main scores of $\rho_m$ (Fig. 2f) and $C_p$ (Fig. 2k) show that values above a certain threshold degrade $\tilde{\rho}$. This effect occurs because both $\rho_m$ and $C_p$ represent heat capacity per volume. A $\rho_m$ or $C_p$ implies a high $C_v$, so the necessary high temperature for melting in the required area cannot be easily achieved. Consequently, if $\rho_m$ or $C_p$ surpasses a certain level, lack of fusion is likely to occur, and this effect decreases $\tilde{\rho}$.

Increase in $C_v$ indicates an increase in the amount of energy that is required to raise the temperature of a certain area (Fig. 3h); as a consequence, as $C_v$ increases, the time required for the temperature to rise around the laser-irradiated area increases. Likewise, decrease in $k$ slows the rate of increase in surrounding temperature. These effects suggest that the interaction between $k$ and $v$ when $k$ is low shows a similar tendency to the interaction between $C_v$ and $v$ when $C_v$ is high. Similarly, when $k$ is high and $C_v$ is low, the interaction between each feature and scan speed displays a similar tendency. In summary, the interactions between $C_v$ and $v$ exhibit opposite trends to the interactions between $k$ and $v$, because the temperature rise behavior around the laser-irradiated area is contrary to each other.

**Prediction of process conditions and experimental validation**

We have developed a method to predict process parameters for a particular powder, which are expected to yield a product that has $\tilde{\rho} \geq 98\%$, by inversely predicting the present model. The goal of this method is to suggest process parameters with the highest likelihood of yielding products that have $\tilde{\rho} \geq 98\%$, as determined by the ML model. We introduced a fitness function

$$\text{Certainty} = (y_{present} - 0.5) \times 2 \tag{3}$$

to evaluate the reliability of predictions made by the present model, which produces prediction values between 0 and 1.

**Table 3 | Process condition prediction results for STS 316 L powder and their respective relative density of the manufactured specimens measured through experiments**

| # | Laser power [W] | Scan speed [mm/s] | Layer thickness [mm] | Hatch spacing [mm] | Relative density [%] |
|---|---|---|---|---|---|
| 1 | 341 | 1000 | 0.05 | 0.103 | 99.9 |
| 2 | 273 | 830 | 0.05 | 0.1 | 99.9 |
| 3 | 234 | 900 | 0.05 | 0.08 | 99.9 |
| 4 | 335 | 860 | 0.05 | 0.148 | 99.9 |
| 5 | 358 | 730 | 0.05 | 0.124 | 99.9 |
| 6 | 340 | 1540 | 0.05 | 0.084 | 99.8 |
| 7 | 395 | 740 | 0.05 | 0.155 | 99.8 |
| 8 | 398 | 1010 | 0.05 | 0.118 | 99.8 |
| 9 | 390 | 650 | 0.05 | 0.188 | 99.5 |
| 10 | 275 | 690 | 0.05 | 0.152 | 99.5 |
| 11 | 141 | 620 | 0.05 | 0.082 | 99.4 |
| 12 | 381 | 1320 | 0.05 | 0.093 | 98.9 |

**Table 4 | Process condition prediction results for AlSi10Mg powder and their respective average relative density of the manufactured specimens measured through experiments**

| # | Laser power [W] | Scan speed [mm/s] | Layer thickness [mm] | Hatch spacing [mm] | Relative density [%] |
|---|---|---|---|---|---|
| 1 | 398 | 2150 | 0.05 | 0.081 | 99.93 |
| 2 | 343 | 1600 | 0.05 | 0.105 | 99.9 |
| 3 | 314 | 1370 | 0.05 | 0.119 | 99.88 |
| 4 | 391 | 1540 | 0.05 | 0.083 | 99.87 |
| 5 | 322 | 1320 | 0.05 | 0.08 | 99.78 |
| 6 | 397 | 1470 | 0.05 | 0.118 | 99.71 |
| 7 | 272 | 1260 | 0.05 | 0.143 | 99.7 |
| 8 | 368 | 1250 | 0.05 | 0.097 | 99.57 |
| 9 | 236 | 990 | 0.05 | 0.131 | 99.48 |
| 10 | 278 | 1230 | 0.05 | 0.098 | 99.43 |
| 11 | 321 | 980 | 0.05 | 0.104 | 99.3 |
| 12 | 362 | 1270 | 0.05 | 0.148 | 98.12 |

To perform inverse prediction of the present model, we performed random search (details in Methods section), which is an advantageous metaheuristic technique to find multiple optimal values. This approach is preferable, because the goal of our method is to suggest several high-certainty process parameters, rather than only the parameter that has the highest certainty. All process condition predictions in this study were conducted with a fixed layer thickness of 0.05 mm, because it was the most suitable for the size of the powder used in this study ($d_{50} = 46$ μm).

The validation was conducted using STS 316 L and AlSi10Mg, which are alloys included in the database used for model training. The process condition prediction and experimental verification for STS 316 L and AlSi10Mg (Tables 3 and 4) serve as tests to verify the model and method's reproducibility. Optical micrographs (Fig. 4a, b) were obtained of the cut surfaces for the six STS 316 L specimens and the six AlSi10Mg specimens that had the highest $\tilde{\rho}$.

All 12 STS 316 L specimens had $\tilde{\rho} \geq 98\%$; 11 had $\tilde{\rho} \geq 99\%$. This result demonstrates the reliability of our model and method. The dataset for training the present model, including the test dataset, contains seven STS 316 L powder data entries[26–28] with process conditions ($141 \leq P \leq 398$ W, $620 \leq v \leq 1540$ mm/s, $t = 0.05$ mm, and $0.08 \leq h \leq 0.188$ mm) (Supplementary Table 1), which are the ranges of process conditions in Table 3. Process condition No. 5 in Table 3 is similar to those in Supplementary Table 1, but the remaining conditions in Table 3 represent new, distinct process conditions. This result illustrates that our model and method effectively reproduce existing optimal processing conditions for the powder in use, and also generate novel optimal processing conditions.

On the other hand, compared to Fe-based alloys like STS 316 L, AlSi10Mg, an Al-based alloy, is more challenging for PBF processing. To get a more accurate reading of its relative density, we prepared multiple samples and averaged the values. The relative densities shown in Table 4 represent these average values. For more detailed information on each process and the exact relative densities, refer to Supplementary Table 2.

The results in Table 4 are promising: all 12 tested process conditions achieved an average $\tilde{\rho}$ over 98%. Impressively, 11 out of these 12 had average $\tilde{\rho} \geq 99\%$. The dataset contains eight data entries[29–31] for AlSi10Mg powder under specific process conditions ($236 \leq P \leq 398$ W, $980 \leq v \leq 2150$ mm/s, $t = 0.05$ mm, and $0.08 \leq h \leq 0.148$ mm) (Supplementary Table 3), which are the ranges of process conditions in Table 4. While process condition No. 9 in Table 4 is similar to those in Supplementary Table 3, the remaining conditions spotlight new and

distinct process parameters. This reaffirms our model's efficacy in both mirroring existing optimal conditions and pioneering novel ones.

To substantiate the transferability of our model and method, we conducted an experimental verification using Fe60Co15Ni15Cr10 MEA, which was not included in the database used to train the model. The prediction results of process conditions for Fe60Co15Ni15Cr10 MEA powder and the experimental verification of these conditions were arranged in order of decreasing $\tilde{\rho}$ of the manufactured specimens (Table 5). Optical micrographs (Fig. 4c) were obtained from the six Fe60Co15Ni15Cr10 MEA specimens had had the highest $\tilde{\rho}$.

To increase the reliability of our experimental results, we fabricated several specimens for each process condition, measured their $\tilde{\rho}$ (Supplementary Table 4), then averaged these values. For all 12 process conditions, specimens made with each process condition exhibited average $\tilde{\rho} \geq 99\%$, which exceeds our established threshold of 98%. This result confirms the reliability and transferability of our model and method, and that they are applicable to generate several optimal process conditions that yield products with high $\tilde{\rho}$, regardless of the material used.

Our experimental results confirm that the method effectively generates process conditions that yield products with high $\tilde{\rho}$ for STS 316 L, AlSi10Mg, and Fe60Co15Ni15Cr10 MEA. However, our method does not assure optimal material properties, such as surface quality, other than $\tilde{\rho}$, because the primary focus of our model and method was to achieve high $\tilde{\rho}$. Consequently, although a specimen may achieve the desired $\tilde{\rho}$ under process conditions recommended by our method, other properties may be suboptimal. In future work, this limitation can be addressed by developing new models for other properties such as surface quality, and incorporating these models into our method.

In summary, we have successfully developed a method that can identify process parameters that yield a product that has $\tilde{\rho} \geq 98\%$, in accordance with the properties of the powder used. This success was made possible by training a model that determines whether a product that was fabricated under specific process conditions and utilizing a particular powder has the $\tilde{\rho} \geq 98\%$. By applying a sigmoid function to transform output, we mitigated bias in the dataset, and thereby increased the accuracy of our ML model. SHAP analysis provided insight into the model's decision-making process, and reproduced well-understood interactions such as that between scan speed and laser power, while also uncovering knowledge regarding the effects of input features or interactions between these features on the $\tilde{\rho}$.

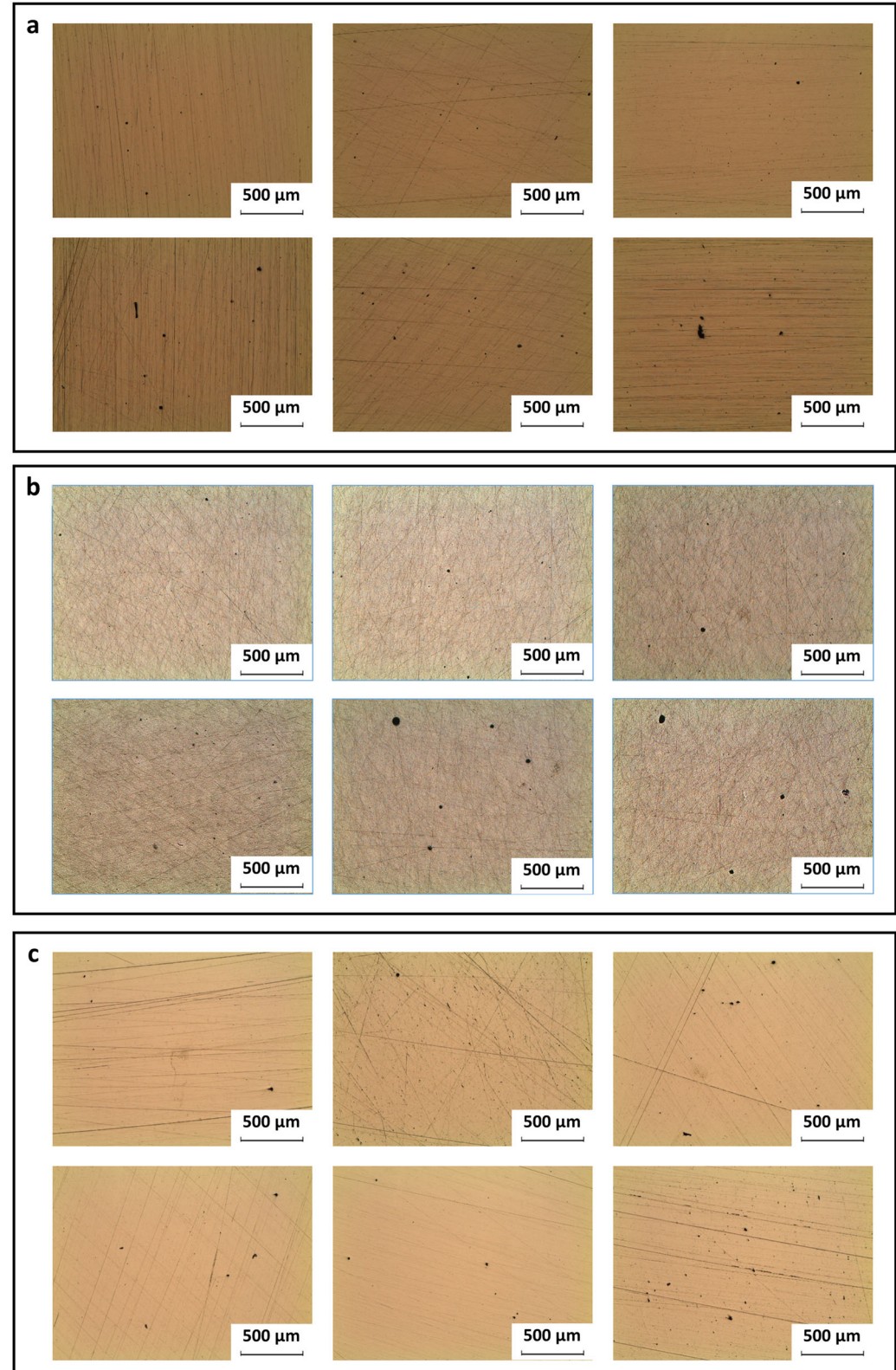

**Fig. 4 | Optical micrographs of the cut surfaces for the specimens with the highest relative density.** Description: Optical micrographs showing the cut surfaces of the top six specimens with the highest relative density for **a** STS 316 L, **b** AlSi10Mg, and **c** Fe60Co15Ni15Cr10 MEA.

Our method, designed to generate process conditions conducive to high $\widetilde{\rho}$, operates by using random search to inversely predict the present model. We substantiated the reproducibility and transferability of our model and method by predicting 12 process conditions for STS 316 L and AlSi10Mg powders of which data were included in the training dataset and Fe60Co15Ni15Cr10 MEA powder of which data were not included in the training dataset, respectively. Experimental validation affirmed that products fabricated using each powder and condition had $\widetilde{\rho} \geq 98\%$, and thus provided empirical evidence that our method can generate process

**Table 5 | Process condition prediction results for Fe60Co15-Ni15Cr10 MEA powder and their respective average relative density of the manufactured specimens measured through experiments**

| # | Laser power [W] | Scan speed [mm/s] | Layer thickness [mm] | Hatch spacing [mm] | Relative density [%] |
|---|---|---|---|---|---|
| 1 | 302 | 980 | 0.05 | 0.103 | 99.97 |
| 2 | 365 | 1010 | 0.05 | 0.118 | 99.90 |
| 3 | 282 | 780 | 0.05 | 0.122 | 99.90 |
| 4 | 349 | 1070 | 0.05 | 0.087 | 99.90 |
| 5 | 367 | 740 | 0.05 | 0.156 | 99.86 |
| 6 | 399 | 840 | 0.05 | 0.102 | 99.86 |
| 7 | 244 | 790 | 0.05 | 0.098 | 99.84 |
| 8 | 397 | 920 | 0.05 | 0.14 | 99.83 |
| 9 | 289 | 540 | 0.05 | 0.151 | 99.78 |
| 10 | 373 | 530 | 0.05 | 0.183 | 99.69 |
| 11 | 392 | 550 | 0.05 | 0.219 | 99.65 |
| 12 | 385 | 1380 | 0.05 | 0.094 | 99.06 |

**Table 6 | Various ML algorithms and the corresponding accuracies achieved by each model**

| Algorithms | Accuracy [%] |
|---|---|
| Random forest | 86.2 |
| XGBoost | 85.8 |
| K-Neighbors neighbors | 81.7 |
| Decision tree | 81.4 |
| Milt-layer perceptron | 80.5 |
| Support vector machine | 79.5 |
| Stochastic gradient descent | 67.8 |
| Ridge | 67.8 |
| Linear regression | 67.8 |
| Ridge CV | 67.7 |
| Bayesian ridge | 67.7 |
| Lasso CV | 67.7 |
| Elastic net CV | 67.7 |
| Gaussian process regression | 66.1 |
| Partial least squared regression | 65.5 |
| Lasso | 51.2 |
| Elastic net | 51.2 |
| Kernel ridge | 41.9 |

conditions that lead to products with high $\widetilde{\rho}$, even for previously unlearned alloys.

## Methods

### Dataset preparation

We collected 2167 process conditions for 50 metal powders and their corresponding $\widetilde{\rho}$ from previous studies (references in supplementary materials) in which products were produced using respective process conditions and powders. The data include powder properties $R$, $k$, $C_p$, $\rho_m$. $T_M$; process conditions $P$, $v$, $t$, $h$; and $\widetilde{\rho}$.

In compiling our dataset, we favored $\widetilde{\rho}$ data measured via image analysis but also included some data measured via the Archimedes method. This approach was adopted for two main reasons: firstly, the discrepancy between the two methods is typically minor within our low-porosity region of interest[32,33], and the errors of relative densities far from 98% are disregarded as the relative density is normalized by the sigmoid function. Secondly, as supported by Halevy et al.[34], larger datasets with some measurement errors are generally more beneficial for training and generalization than smaller, flawless datasets. It is important to note that that even data measured solely by image analysis might not be entirely free from experimental errors.

While other conditions including machine type and laser diameter could potentially serve as meaningful input features, we consciously decided to limit the number of input features due to the size of our dataset. Adding more input features would increase the dimensionality of the dataset's input space, which, without a corresponding increase in data quantity, can lead to the "curse of dimensionality"[35]. This phenomenon can hinder effective learning and decrease the interpretability of the model[36]. As such, we focused on major process conditions, $P$, $v$, $t$, and $h$, as our model's input features.

Our data compilation strategy ensures the dataset is complete, with no missing values. The powder properties were predominantly sourced from existing literature. When direct collection was not feasible, we computed the properties, as we will discuss in further detail later. The process conditions, $P$, $v$, $t$, and $h$, are typically reported in most PBF-related research publications. We ensured data quality by excluding any studies that did not provide these crucial details, thus maintaining the integrity of our dataset.

As we mentioned above, most powder properties were obtained from the literature[37–54]. But for some powders, $C_p$ and $T_M$ were not available, so we estimated them by thermodynamic calculation using the TCFE2000 thermodynamic database and its upgraded version[55–59]

with Thermo-Calc software[60]. $R$ of almost all alloy powders could not be found in the literature, we calculated $R$ as a weighted arithmetic average of the composition, assuming that the alloy powder reflectivity followed the rule of mixture. $R$ of pure elements for laser with a wavelength of 1 μm was also collected from literature[61–68]. For Fe60Co15Ni15Cr10 MEA, we calculated its properties or measured them experimentally. For MEA, we used thermodynamic calculation to determine its $T_M = 1461.75 °C$, and computed its $R = 66.15\%$ as the weighted arithmetic average of the composition.

We further calculated the thermal conductivity of the MEA as

$$k = \alpha \rho_m C_p = 13.66 \text{W}/(\text{m} \cdot \text{K}), \qquad (4)$$

where $\alpha = 3.739$ mm²/s is thermal diffusivity, as measured using the laser flash technique (ASTM E1461, LFA 467, NETZSCH, Germany), $\rho_m = 7.91$ g/cm³ was determined using Archimedes' principle (XP205, Mettler Toledo, USA), and $C_p = 0.462$ J/(g · K) was measured using a Differential Scanning Calorimeter (DSC, DSC8000, PerkinElmer, USA). The collected dataset is provided in Supplementary Data 1.

### ML Model

In this study, we aimed to develop the most accurate model to address our problem; for this purpose, we selected the most suitable among a range of ML algorithms. We trained regression models with sigmoid-transformed outputs using several algorithms, all operating on the same training and test datasets. The highest test set accuracies were obtained using the Random Forest (86.2%) and XGBoost (85.8%) algorithms (Table 6).

To further improve these results, we optimized the hyperparameters of the Random Forest and XGBoost models, by using the hyperparameter optimization framework, Optuna[69]. We then trained 100 versions of each model, with shuffled training, validation, and test datasets. Comparing the average test set accuracy of the models, the Random Forest model achieved an accuracy of 85.7% and XGBoost achieved 87%. Consequently, we selected the XGBoost algorithm to train our final model. More details about the XGBoost model and the others are in the supplementary materials.

The XGBoost model was developed using the XGBoost library[19], an open-source software library providing a gradient boosting

framework. All other models were developed using scikit-learn[70] a free software ML library for Python. The input data for all models was standardized using mean and standard deviation values. To accurately evaluate the models' performances, we applied the holdout validation method. This technique randomly partitions the dataset into training, validation, and test sets. In our application, we allocated 70% of the data for training purposes, while the remaining 30% was equally partitioned between validation and testing, each constituting 15%.

Our primary model is an XGBoost model, which uses a tree structure, so we used tree SHAP (tree explainer)[24] to calculate the SHAP[23] scores. We analyzed our model by extracting the SHAP main scores and SHAP interaction scores by using the tree explainer function, which enables calculation of SHAP interaction values.

### Method for prediction of process conditions

The method detailed here outlines how to identify process conditions that are expected to yield products that have $\widetilde{\rho} \geq 98\%$, considering the specific powder used, by inverse prediction of the present model. This technique incorporates a form of random search that involves three primary steps: random input generation, prediction, and selection.

The process initiates with the input of powder properties and the determination of process parameters, including which are fixed, and their values. Then the random-input-generation step produces even number of sets of inputs within the range of existing data for each input in the dataset. Subsequently, during the prediction step, these sets of inputs are standardized using the mean and standard deviation used during training, then fed into the model that had been trained to predict $\widetilde{\rho}$.

The present model outputs values between 0 and 1, which are relative densities transformed by the sigmoid function, so they can be converted to Certainty (Eq. 3). The following step selects the half of the input sets that have the highest predicted value. However, any input sets that have process conditions that are deemed too similar to others are discarded before this selection process. The criterion for similarity is defined as:

$$\sum |a_i - b_i| < 1, \tag{5}$$

where $a_i$ and $b_i$ represent the standardized process parameters of two input sets.

To replace discarded sets, new random input sets are generated. By repeating this process, we can identify process conditions that are expected to yield products that have high $\widetilde{\rho}$. The iteration stops when the sets of inputs selected remain consistent for a certain number of iterations.

This method to predicts optimal process conditions considering the powder used, has been incorporated into a GUI program. This program, along with its user manual, is available for download at https://doi.org/10.5281/zenodo.8382890[71]. For optimal printing results, we recommend using a layer thickness that exceeds the $d_{50}$ size of the powder used in the process.

### Experimental procedure

To evaluate the accuracy of our prediction method, we printed coupon samples of 316 L stainless steel, AlSi10Mg, and Fe60Co15Ni15Cr10 MEA, then measured their porosities. The metallic powder used for the L-PBF process was gas-atomized spherical powder with a size distribution of $d_{50} = 46$ μm (MK Inc., Republic of Korea). Using a commercial L-PBF machine (Concept Laser M2, GE Additive, USA), we printed cubic coupons of dimensions $10 \times 10 \times 10$ mm³. Process parameters were selected for model validation (Tables 3, 5). Other process parameters: layer thickness of 50 μm, a laser spot size of 50 μm, a layer-by-layer rotation angle of 90°, and printing in $N_2$ atmosphere, were maintained across all experiments.

The porosity of the samples was analyzed from the XZ-plane (where X denotes the powder coating direction and Z represents the building direction). The coupons were cut along this plane and subsequently polished mechanically to a 1200 mesh with emery paper. We then captured optical microscopic images of the polished sections and used ImageJ software to analyze and measure the porosity.

### Reporting summary

Further information on research design is available in the Nature Portfolio Reporting Summary linked to this article.

### Data availability

The raw data generated in this study are available within the article and its supplementary materials. The same dataset has been deposited in the Figshare repository under the https://doi.org/10.6084/m9.figshare.24203799[72]. There are no restrictions on data access. Source data are provided with this paper.

### Code availability

The codes generated in this study, along with the GUI program designed to predict process conditions for various metallic powders, are available for download at https://doi.org/10.5281/zenodo.8382890.[71]

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

## Acknowledgements

This work has been financially supported by the National Research Foundation of Korea (NRF) funded by the Ministry of Science and ICT, Korea (Grant No. NRF-2022R1A5A1030054 to B.-J.L. and Grant No. NRF-2022R1A2C2004331 to B.-J.L.).

## Author contributions

J.W.: conceptualization, methodology, software, validation, formal analysis, investigation, data curation, writing—original draft, visualization. S.G.J.: writing—review and editing, validation, formal analysis, investigation, resources. E.S.K.: conceptualization, investigation, resources. H.S.K.: supervision, project administration, funding acquisition. B.-J.L.: conceptualization, methodology, writing—review and editing, supervision, project administration, funding acquisition.

## Competing interests

The authors declare no competing interests.
