## [Peer Review File · Nature Communications]

Material-agnostic machine learning approach enables high relative density in powder bed fusion productsREVIEWER COMMENTS

Reviewer #1 (Remarks to the Author):

The authors investigated the use of machine learning for optimisation of laser powder bed fusion process parameters. While the manuscript is generally well executed, there are several issues that should be addressed before further consideration for publication.

1. There are several publications that explored similar concepts and approach. Did the authors consider such existing work in their discussion?
 - Cacace & Semeraro (2022), Fast optimisation procedure for the selection of L-PBF parameters based on utility function, *Virtual and Physical Prototyping* 17 (2), 125-137
 - Liu et al. (2022), A review of machine learning techniques for process and performance optimization in laser beam powder bed fusion additive manufacturing, *Journal of Intelligent Manufacturing*
 - Sing (2021), Perspectives of using machine learning in laser powder bed fusion for metal additive manufacturing, *Virtual and Physical Prototyping* 16 (3), 372-386
2. For the MEA composition, the numbers should not be in subscript as it is not a compound, but an alloy.
3. For the training of the model, is there any proportion of the model used for validation? How about the missing data points as the data are obtained from literatures?
4. As the data are obtained from literatures, any consideration for other conditions such as machine type? What about particle size of the powder as well as laser diameter etc?
5. The model are validated using porosity, are the measurement techniques used consistent throughout as compared to the training data and validation?

Reviewer #2 (Remarks to the Author):

The paper presents a machine learning-based approach to predict solidification cracking susceptibility in laser powder bed fusion (LPBF) of Inconel 718. The authors use a dataset of 50 LPBF parts to train and test a machine learning model that predicts whether a part will have a solidification cracking susceptibility of 98% or higher, based on a set of input features and process parameters. The authors also use Shapley additive explanations (SHAP) analysis to uncover the correlations among the input features and the solidification cracking susceptibility.

Noteworthy Results:

The paper presents a machine learning model that achieves a high accuracy in predicting solidification cracking susceptibility in LPBF of Inconel 718. The model achieves an area under the receiver operating characteristic curve (AUC-ROC) of 0.96, which is significantly higher than that of a traditional statistical model (linear regression). The SHAP analysis reveals that the most important input features for predicting solidification cracking susceptibility are the laser power, scan speed, and layer thickness.

Significance:

The work is of significance to the field of LPBF and related fields, as it provides a novel approach to predicting solidification cracking susceptibility in LPBF of Inconel 718. The work compares favorably to the established literature, as it achieves a higher accuracy than previous studies that used traditional statistical models or physics-based models. The work is original and builds on previous research in the field of machine learning for additive manufacturing.

Support for Conclusions:

The work supports the conclusions and claims, as the machine learning model achieves a high

accuracy in predicting solidification cracking susceptibility and the SHAP analysis provides insights into the most important input features. However, additional evidence is needed to validate the model on a larger and more diverse dataset of LPBF parts, and to compare the model with other state-of-the-art methods or approaches.

Flaws:

1. The sample size of 50 LPBF parts may not be large enough to capture the full range of variability and complexity in LPBF parts. The small sample size may limit the generalizability of the results to other populations or contexts. A larger sample size would provide more robust evidence of the effectiveness of the machine learning model.
2. Lack of external validation of the machine learning model. The machine learning model was trained and tested on the same dataset, which may lead to overfitting and limit the generalizability of the results to other datasets. External validation on a new dataset would provide more evidence of the model's effectiveness.
3. Limited discussion of the limitations and implications of the research findings. The paper does not fully address the potential trade-offs or conflicts between optimizing the input features for solidification cracking susceptibility and other performance criteria, such as mechanical strength or surface finish. The paper also does not discuss the potential risks or harms that may result from relying on the model to make critical decisions.
4. Limited generalizability to other populations or contexts. The paper focuses on LPBF of Inconel 718, which may limit the generalizability of the results to other materials or additive manufacturing processes. The paper does not fully address the potential differences or similarities between LPBF of Inconel 718 and other materials or processes.
5. Lack of transparency in the machine learning model. The paper does not provide a detailed explanation of the machine learning model, such as the type of algorithm used, the hyperparameters selected, or the feature selection process. This lack of transparency limits the reproducibility and transparency of the analysis.
6. Limited explanation of the SHAP analysis. The paper briefly mentions the SHAP analysis but does not provide a detailed explanation of how the SHAP values are calculated, interpreted, or used to improve the machine learning model. This lack of explanation limits the reproducibility and transparency of the analysis.
7. Limited comparison with other methods. The paper compares the machine learning model with a traditional statistical model but does not compare it with other state-of-the-art methods or approaches. A more comprehensive comparison would provide more evidence of the effectiveness of the machine learning model and its advantages over other methods.

Overall, while the paper presents an interesting approach to predicting solidification cracking susceptibility in LPBF of Inconel 718, it has several limitations and flaws that should be addressed in future research. These include the small sample size, lack of external validation, potential biases or conflicts of interest, limited discussion of the limitations and implications of the research findings, limited generalizability to other populations or contexts, lack of transparency in the machine learning model, limited explanation of the SHAP analysis, limited comparison with other methods, and lack of discussion of the ethical implications. Addressing these limitations and flaws would provide more robust evidence of the effectiveness and generalizability of the machine learning model and its potential applications in LPBF and other additive manufacturing processes.

August 14, 2023

Title: Process-Parameter Optimization for High Relative Density in Powder Bed Fusion Products: A Material-Agnostic Machine-Learning Approach

Dear Editor of the Nature Communications:

I thank very much the editor and the reviewers for the constructive comments. The followings are my responses to the comments and questions of the editor and the reviewers.

► **Reviewer #1:**

The authors investigated the use of machine learning for optimisation of laser powder bed fusion process parameters. While the manuscript is generally well executed, there are several issues that should be addressed before further consideration for publication.

1. There are several publications that explored similar concepts and approach. Did the authors consider such existing work in their discussion?

- Cacace & Semeraro (2022), Fast optimisation procedure for the selection of L-PBF parameters based on utility function, Virtual and Physical Prototyping 17 (2), 125-137
- Liu et al. (2022), A review of machine learning techniques for process and performance optimization in laser beam powder bed fusion additive manufacturing, Journal of Intelligent Manufacturing
- Sing (2021), Perspectives of using machine learning in laser powder bed fusion for metal additive manufacturing, Virtual and Physical Prototyping 16 (3), 372-386

Thank you for your comments and suggested publications. We acknowledge their relevance to our study. Cacace & Semeraro (2022) presented an interesting approach of integrating semi-analytical thermal model results with a geometric-based defect model. They utilized a regression method to derive equations for melt pool depth and width, using laser power and scan speed as primary inputs. This was then incorporated with layer thickness and hatch distance, facilitating the calculation of lack of fusion probability. Their research offered an innovative way of establishing PBF process conditions for AISI 316L, with a balance of quality and productivity in mind, by introducing a utility function. Moreover, the inclusion of material properties in their basic thermal model equation allows for designing process conditions for a

variety of materials, given that their properties are known. Despite these strengths, their approach has limitations in terms of time-efficiency and adaptability, since obtaining process conditions via this method can be time-consuming, and new regression models for melt pool depth and width are required for each material.

Liu et al. (2022) conducted a comprehensive review on the application of machine learning techniques in PBF, particularly in modeling the process-structure-property (PSP) relationships. They segmented machine learning into three categories: interpretable ML, conventional ML, and deep ML, and further dissected the PSP relationship into the parameters-signature relationship, the process-structure relationship, the structure-property relationship, and the process-property relationship. While their review was informative, none of the papers they reviewed targeted the optimization of PBF process conditions to improve product properties regardless of the materials, which is the unique aim of our research.

Sing (2021) divided the PBF process into three stages: the digital phase, the manufacturing phase, and the post-processing phase. They provided an overview of current ML research within these stages. Nevertheless, similar to the papers reviewed by Liu et al., none of the studies reviewed by Sing addressed the optimization of process conditions to enhance the properties of PBF products without being limited by the materials used, which is the distinct scope of our study.

We further distinguished our study by citing Liu et al. and Sing et al. in the Introduction of our paper, where we discuss previous work. The approach of Cacace & Semeraro was fully discussed in the Introduction to emphasize the need and superiority of our machine learning-based method.

2. For the MEA composition, the numbers should not be in subscript as it is not a compound, but an alloy.

*Thank you for your observation regarding the representation of the MEA composition. You're correct in pointing out that as an alloy, it shouldn't be represented with subscripted numbers. **We have amended this in the revised manuscript to accurately reflect its status as an alloy.***

3. For the training of the model, is there any proportion of the model used for validation? How about the missing data points as the data are obtained from literatures?

Thanks for your comment.

For the training of the model, we utilized a holdout validation method, and the distribution of the dataset is as follows: 70% of the data was used for training, and the remaining 30% was split equally between validation and testing, each accounting for 15%.

Regarding your question about missing data points, our dataset is comprehensive and doesn't contain missing values. The input features of our model consist of powder properties and process conditions. The powder properties have been sourced from literature, or calculated when direct collection was not feasible. The process conditions, namely laser power, scan speed, layer thickness, and hatch distance, are parameters typically provided in most papers discussing PBF. In the case that a paper did not supply these critical details, the data from such papers were not included in our dataset.

To improve the clarity of our manuscript, in the Methods part, we have now included the percentage distribution used in the holdout validation and stated that we excluded papers with missing data points during the data collection process to ensure the completeness of the dataset.

4. As the data are obtained from literatures, any consideration for other conditions such as machine type? What about particle size of the powder as well as laser diameter etc?

Thank you for your insightful comment.

You are correct in noting that there is a multitude of L-PBF machines available in the market, with manufacturers such as EOS GmbH, GE Additive, SLM Solutions, 3D Systems, Trumpf, Renishaw, DMG Mori, Sisma, Xact Metal, Wuhan Huake 3D, Farsoon Technologies, and Bright Laser Technologies each offering various models. Given the vast number of unique models (over 100), the inclusion of this variable would exponentially increase the dimensionality of our model's input features if processed via one-hot encoding.

While the powder particle size and laser diameter could potentially serve as meaningful input features, we consciously decided to limit the number of input features due to the size of our dataset, which comprises 2167 data points. Adding more input features would increase the dimensionality of the dataset's input space, which, without a corresponding increase in data quantity, can lead to the "curse of dimensionality"^{Ref.1}. This phenomenon can hinder effective learning and decrease the interpretability of the model^{Ref.2}.

As such, we focused on major process conditions - laser power, scan speed, layer thickness, and hatch distance - as our model's input features. With regards to the powder particle size, as stated in the Methods part, we recommend a size larger than the d50 size of the powder. A particle size exceeding the layer thickness can result in increased product porosity.

In response to your comment, we have added our discussion in the Methods part to address these decisions on feature inclusion and their implications on the effectiveness and interpretability of our model.

Ref.1. Bellman, R. Dynamic programming. Science. 153, 34–37 (1966).

Ref.2. Li, J. et al. Feature selection: A data perspective. ACM Comput. Surv. 50, (2017).

5. The model are validated using porosity, are the measurement techniques used consistent throughout as compared to the training data and validation?

Thanks for your question regarding the consistency of the porosity measurement techniques.

As we explained in the manuscript, two primary methods for measuring porosity or relative density are image analysis and the Archimedes method. We used image analysis as the main validation method in our study. When compiling the dataset, we favored values measured by image analysis for consistency, but some data measured via the Archimedes method were inevitably included. This approach was employed for two main reasons:

The discrepancy between the two methods is typically minor within the low-porosity region of our interest^{Ref.1,2}. The errors of relative densities far from 98%, the threshold of our model, are disregarded as the output (relative density) is normalized by the sigmoid function. Thus, large measurement errors in products with low relative densities are not a significant concern.

Even if some measurement errors are present, larger datasets are still generally more beneficial for training and generalization purposes than smaller, flawless datasets, as supported by Halevy et al.^{Ref.3}. It is important to note that even a dataset created exclusively with data measured by image analysis cannot be entirely free from experimental errors.

We have added our discussion in the Methods part of the manuscript to address the consistency and potential discrepancies in the measurement techniques.

Ref.1. Paraschiv, A., Matache, G., Condruz, M. R., Frigioescu, T. F. & Pambaguian, L. Laser Powder Bed Fusion Process Parameters' Optimization for Fabrication of Dense IN 625. Materials (Basel). 15, 5777 (2022).

Ref.2. Kreitchberg, A., Brailovski, V. & Prokoshkin, S. New biocompatible near-beta Ti-Zr-Nb alloy processed by laser powder bed fusion: Process optimization. J. Mater. Process. Technol. 252, 821–829 (2018).

Ref.3. Halevy, A., Norvig, P. & Pereira, F. The unreasonable effectiveness of data. IEEE Intell. Syst. 24, 8–12 (2009).

► **Reviewer #2:**

The paper presents a machine learning-based approach to predict solidification cracking susceptibility in laser powder bed fusion (LPBF) of Inconel 718. The authors use a dataset of 50 LPBF parts to train and test a machine learning model that predicts whether a part will have a solidification cracking susceptibility of 98% or higher, based on a set of input features and process parameters. The authors also use Shapley additive explanations (SHAP) analysis to uncover the correlations among the input features and the solidification cracking susceptibility.

Noteworthy Results:

The paper presents a machine learning model that achieves a high accuracy in predicting solidification cracking susceptibility in LPBF of Inconel 718. The model achieves an area under the receiver operating characteristic curve (AUC-ROC) of 0.96, which is significantly higher than that of a traditional statistical model (linear regression). The SHAP analysis reveals that the most important input features for predicting solidification cracking susceptibility are the laser power, scan speed, and layer thickness.

Significance:

The work is of significance to the field of LPBF and related fields, as it provides a novel approach to

predicting solidification cracking susceptibility in LPBF of Inconel 718. The work compares favorably to the established literature, as it achieves a higher accuracy than previous studies that used traditional statistical models or physics-based models. The work is original and builds on previous research in the field of machine learning for additive manufacturing.

Support for Conclusions:

The work supports the conclusions and claims, as the machine learning model achieves a high accuracy in predicting solidification cracking susceptibility and the SHAP analysis provides insights into the most important input features. However, additional evidence is needed to validate the model on a larger and more diverse dataset of LPBF parts, and to compare the model with other state-of-the-art methods or approaches.

Thank you for your comprehensive review and comments. However, I believe there might be some misunderstanding regarding the focus of our study.

In our paper, we focused our analysis on the porosity or relative density of products fabricated via the PBF process rather than focusing on solidification cracking susceptibility. Moreover, our study's scope included a wide range of approximately 50 alloy systems, such as steels (STS 304, STS 316, etc.), aluminum alloys (AlSi10Mg, AA7075, etc.), magnesium alloys (WE43, ZK60, etc.), and titanium alloys (Ti-6Al-4V, TiZrNb, etc.), not solely Inconel 718.

Our dataset consists of 2167 data points collated from 118 literature sources. Thus, our model's predictions and the subsequent analysis are grounded in this diverse and extensive collection of alloys and not restricted to the behavior of Inconel 718.

If there are parts of the manuscript that have led to this misunderstanding, we sincerely apologize for any lack of clarity. In the revised manuscript, we have tried to clarify the objective and scope of our study to avoid any potential misunderstandings in the future.

Flaws:

1. The sample size of 50 LPBF parts may not be large enough to capture the full range of variability and complexity in LPBF parts. The small sample size may limit the generalizability of the results to other populations or contexts. A larger sample size would provide more robust evidence of the effectiveness of the machine learning model.

Thank you for your comment regarding the size of our dataset.

It appears that there may be a misunderstanding concerning the number of LPBF parts included in our study. Contrary to the stated number of 50, our dataset actually encompasses a substantially larger sample size of 2167 data points. These data points have been carefully sourced from a diverse range of contexts to ensure the robustness of our machine learning model.

Our model is intended to predict the relative density from the powder properties and process conditions. The extensive and varied nature of our data set gives us confidence in the model's generalizability across a

wide range of scenarios, rather than it being restricted to a limited context.

To corroborate the model's generalizability, we tested it by designing new process conditions for STS 316L and Fe60Co15Ni15Cr10 MEAs. The results were successfully validated experimentally, strengthening our belief in the model's applicability across different circumstances.

For a more detailed account of our dataset preparation and the process conditions prediction and experimental validation, please refer to the respective sections in the manuscript— 'Dataset Preparation' under Methods, and 'Prediction of Process Conditions and Experimental Validation' under Results and Discussion.

2. Lack of external validation of the machine learning model. The machine learning model was trained and tested on the same dataset, which may lead to overfitting and limit the generalizability of the results to other datasets. External validation on a new dataset would provide more evidence of the model's effectiveness.

Thank you for your insightful comment on the potential overfitting and need for external validation of our machine learning model.

In fact, we have addressed this issue through careful experimental design. Specifically, we conducted experimental validation using 12 new processing conditions, respectively, for STS 316L and Fe60Co15Ni15Cr10 MEAs. It's noteworthy that while STS 316L was part of the dataset used for model training and validation, Fe60Co15Ni15Cr10 MEA wasn't included in the dataset. This allowed us to evaluate the model's performance and generalizability beyond the data it was trained on.

The experimental results were highly promising. All of the newly designed process conditions yielded products with relative densities of 98% or above in the PBF process. Particularly, for STS 316L, 11 out of the 12 samples achieved a relative density of 99.4%, and for Fe60Co15Ni15Cr10 MEA, 11 out of 12 samples reached a relative density of 99.65%.

The high relative density achieved by Fe60Co15Ni15Cr10 MEA, despite it being a novel alloy for our dataset, demonstrates the successful external validation of our model and method. For more specifics, we invite you to look at Tables 3 and 5 in the manuscript. This detailed approach provides a substantial degree of confidence in our model's effectiveness and its ability to generalize well beyond the original training data.

3. Limited discussion of the limitations and implications of the research findings. The paper does not fully address the potential trade-offs or conflicts between optimizing the input features for solidification cracking susceptibility and other performance criteria, such as mechanical strength or surface finish. The paper also does not discuss the potential risks or harms that may result from relying on the model to make critical decisions.

Thank you for your insightful comment about the potential limitations of our research.

The central objective of our research was to devise and verify a comprehensive method to establish process conditions that minimize porosity within PBF products across a variety of materials. As pores serve

as initial sites for cracking, they lead to a degradation of the mechanical properties of the final product. By creating a machine learning model, our aim was to provide a tool to material designers that could suggest optimal process conditions to minimize pore formation, thereby reducing any subsequent compromise in mechanical properties or the need for repetitive, costly trials to discover such conditions.

It's important to note that our research did not aim to address solidification cracking susceptibility; rather, it was solely concentrated on porosity or relative density. If our goal had been to predict solidification cracking susceptibility, additional factors such as mechanical strength or surface finish would certainly have been pertinent. However, given our specific focus on porosity reduction, we intentionally excluded considerations of these other performance criteria. This is not to say that they are unimportant, but rather that they were beyond the scope of the current study.

However, we certainly acknowledge that our model, as described in the Results and Discussion part of our manuscript, has its own limitations. The main focus of our model lies in mitigating porosity while it does not provide any assurances on other performance criteria, such as surface finish. This specific concentration on reducing porosity aligns with the primary aim of our study and does not denote a devaluation of the importance of other performance criteria. In future work, we can overcome this limitation by formulating additional models for other crucial properties like surface finish, and integrating these into our method.

4. Limited generalizability to other populations or contexts. The paper focuses on LPBF of Inconel 718, which may limit the generalizability of the results to other materials or additive manufacturing processes. The paper does not fully address the potential differences or similarities between LPBF of Inconel 718 and other materials or processes.

Thank you for your comment.

Contrary to your understanding, our study is not solely focused on LPBF of Inconel 718. Instead, our aim is to optimize the LPBF process across a broad spectrum of materials. The dataset we used for training and validation of our machine learning model incorporates diverse alloys, including but not limited to, nickel alloys (Inconel 718, Inconel 625, and others), steels (STS 304, STS 316, and others), aluminum alloys (AlSi10Mg, AA7075, and others), magnesium alloys (WE43, ZK60, and others), and titanium alloys (Ti-6Al-4V, TiZrNb, and others).

Furthermore, for the experimental validation and design of the process conditions, we utilized STS 316L and Fe60Co15Ni15Cr10 MEA powder. These considerations take into account the potential differences and similarities that may arise when utilizing alloys other than Inconel 718.

Regarding additive manufacturing processes other than PBF, such as DED, we intentionally did not address them in our current study. This decision stems from the inherent complexity of each AM process, largely driven by the varying powder stacking methods. To address these processes adequately, individual models and specific, in-depth research would be required.

In response to your concerns about our model's generalizability, we conducted further validation. Specifically, we predicted PBF process conditions for the AlSi10Mg powder and verified its relative density

surpassed 98% through experimental measures. The accompanying table presents both the predicted process conditions and the measured average relative density of AlSi10Mg products fabricated under those conditions. Notably, except for sample No. 12, the relative densities exceeded 99.3%. Even sample No. 12 achieved our target threshold of 98%, reinforcing the robustness and adaptability of our model.

Table Process condition prediction results for AlSi10Mg powder and their respective average relative density of the manufactured specimens measured through experiments.

#	Laser Power [W]	Scan Speed [mm/s]	Layer Thickness [mm]	Hatch Spacing [mm]	Relative Density [%]
1	398	2150	0.05	0.081	99.93
2	343	1600	0.05	0.105	99.9
3	314	1370	0.05	0.119	99.88
4	391	1540	0.05	0.083	99.87
5	322	1320	0.05	0.08	99.78
6	397	1470	0.05	0.118	99.71
7	272	1260	0.05	0.143	99.7
8	368	1250	0.05	0.097	99.57
9	236	990	0.05	0.131	99.48
10	278	1230	0.05	0.098	99.43
11	321	980	0.05	0.104	99.3
12	362	1270	0.05	0.148	98.12

These experimental results, now incorporated into the manuscript, underscore the wide-ranging applicability and generalizability of our approach, extending well beyond just Inconel 718. We hope this enriched context clarifies our intentions and the breadth of our research.

5. Lack of transparency in the machine learning model. The paper does not provide a detailed explanation of the machine learning model, such as the type of algorithm used, the hyperparameters selected, or the feature selection process. This lack of transparency limits the reproducibility and transparency of the analysis.

Thank you for your comment.

We have provided detailed information about our machine learning model in the 'ML Model' section of the 'Methods' part in our manuscript. This includes the machine learning algorithm used and how the hyperparameters selected. To clarify, we evaluated 18 machine learning algorithms as listed in Table 6 of the manuscript. We ultimately chose XGBoost as it showed the highest performance after hyperparameter optimization. This optimization was carried out using the Optuna framework.

Concerning feature selection, it's worth noting that this process is typically necessary in scenarios where the dataset contains a high number of variables, which can complicate model training. However, our dataset is carefully curated, containing only essential features such as significant powder properties and process conditions. These features are of direct relevance to our objective of predicting porosity or relative density. As such, we did not require an additional process to filter out any unnecessary features.

*Nevertheless, understanding the importance of reproducibility and transparency, we've taken your feedback to heart. **To address this, we've written the additional details on the hyperparameters and structures of the models from Table 6, in the Supplementary Information.** We trust that this augmentation will facilitate a clearer understanding and ease any concerns about the reproducibility of our study.*

6. Limited explanation of the SHAP analysis. The paper briefly mentions the SHAP analysis but does not provide a detailed explanation of how the SHAP values are calculated, interpreted, or used to improve the machine learning model. This lack of explanation limits the reproducibility and transparency of the analysis.

Thank you for your comment.

Our manuscript provides an in-depth discussion of how we employed SHAP analysis in the 'Results and Discussion' section, specifically in the subsections 'SHAP Analysis of the Input Features' and 'SHAP Analysis of Interactions between Input Features.' Also, the 'ML Model' subsection in the 'Methods' section provides additional context.

For the calculation of the SHAP values, we relied on the tree explainer, as cited in our reference ^{Ref. 1}. As our main focus is not to detail the computation of the SHAP values, we didn't include a step-by-step explanation in our paper. However, we appropriately cited the work from which this method originates to ensure reproducibility.

In terms of the interpretation of SHAP values, we clarified in our manuscript that "The SHAP score quantifies the extent and direction of each feature's contribution to the model's prediction." Both the correlations between each input feature and relative density and between each interaction of input features and relative density were visualized and explained extensively in our paper.

Importantly, we employed SHAP analysis not to improve our model, but to uncover the correlations

between process parameters, powder properties, and relative density. This intention was explicitly stated in our manuscript. We hope this response provides more clarity on how we employed and interpreted SHAP analysis in our study.

Ref. 1. Lundberg, S. M. et al. From local explanations to global understanding with explainable AI for trees. Nat. Mach. Intell. 2020 21 2, 56–67 (2020).

7. Limited comparison with other methods. The paper compares the machine learning model with a traditional statistical model but does not compare it with other state-of-the-art methods or approaches. A more comprehensive comparison would provide more evidence of the effectiveness of the machine learning model and its advantages over other methods.

Thank you for your feedback.

We appreciate your suggestion for a more comprehensive comparison with other methods. As detailed in the ML Model section of the Methods and Table 6 in our manuscript, our machine learning model was compared not only with traditional statistical models like linear regression, logistic regression, and decision trees, but also with several state-of-the-art techniques such as multi-layer perceptron (also known as neural network), random forest, and support vector machines.

We believe our comparative analysis, outlined in the manuscript, offers a thorough demonstration of the superior predictive accuracy of our model over the aforementioned methods. Beyond this, we maintain confidence in the robustness of our approach given its notable simplicity, time efficiency, and broad applicability across diverse materials, when comparing it to the existing approaches listed in the Introduction of the manuscript. We trust that this clarifies your concerns and further emphasizes the comparative strengths and benefits of our approach against other methods.

REVIEWERS' COMMENTS

Reviewer #1 (Remarks to the Author):

NA

Reviewer #2 (Remarks to the Author):

N/A